# The Effect of Offspring Gender Composition on Modern Contraceptive Uptake Among Married Women of Reproductive Age in Pakistan: A Facility-Based Cross-Sectional Study

**DOI:** 10.3390/ijerph22010118

**Published:** 2025-01-17

**Authors:** Muhammad Ishaque, Jack Hazerjian, Mohamad Ibrahim Brooks, Tabinda Sarosh, Madiha Latif, Maisam Ali

**Affiliations:** 1Pathfinder Pakistan, Khayaban-e-Jami, Block 9 Clifton, Karachi 75600, Sindh, Pakistan; 2Pathfinder International, 1015 15th Street NW, Suite 1100, Washington, DC 20005, USAtsarosh@pathfinder.org (T.S.)

**Keywords:** gender norms, modern contraceptive uptake, family planning, reproductive health, Pakistan

## Abstract

**Introduction:** Pakistan is confronted with the formidable challenge of high population growth, which is compounded by cultural norms that prioritize male offspring, leading to adverse implications for family planning efforts and demographic trends. Despite efforts to promote contraception, including a national family planning program, Pakistan continues to struggle with low and stagnant contraceptive prevalence rates among married women. The influence of gender composition on modern contraceptive uptake remains underexplored, necessitating research to elucidate its impact on reproductive behavior. **Materials and methods:** This study used the dataset of a facility-based cross-sectional survey conducted in six districts of the Sindh and Punjab provinces in Pakistan. A subset of 495 married women of reproductive age seeking health services from March to June 2019 was used for this study. Logistic regression analysis was employed to examine the association between the gender composition of children and modern contraceptive uptake, adjusting for covariates such as province, the age of the women, and the type of health facility. **Results:** The analysis revealed a significant association between the gender composition of children and modern contraceptive uptake among married women. As the number of daughters increased without sons, the likelihood of contraceptive uptake remained low (adjusted odds ratio [AOR]: 0.12; 95% CI: 0.04–0.34; *p* < 0.000), while having at least one son substantially increased the odds of contraceptive use (AOR: 19.91; 95% CI: 8.00–49.50; *p* < 0.000). Notably, the gender composition of having one daughter with two sons had the highest level of contraceptive uptake, potentially because of family composition preferences. **Discussion:** The findings highlight the pervasive influence of gender composition on reproductive decision-making in Pakistan, with a clear preference for sons driving modern contraceptive behavior. These results underscore the need for targeted interventions to address gender norms and biases while promoting equitable access to family planning services. Engaging men in family planning initiatives is crucial for challenging traditional gender norms and fostering informed decision-making regarding contraception. **Conclusions:** Gender preference influences modern contraceptive uptake among women in Pakistan, with the strong preference for sons driving reproductive behavior. Addressing gender norms and biases while promoting informed, self-determined choice is essential for enhancing modern contraceptive uptake and achieving sustainable population growth. Targeted interventions, including male engagement strategies, are needed to challenge societal gender norms and empower individuals to make autonomous decisions regarding family planning.

## 1. Introduction

Pakistan, the sixth most populous country globally, faces substantial challenges due to its high population growth rate of 2.55% annually, resulting in a population of 241.49 million [1]. The latest Pakistan Demographic and Health Survey (PDHS) 2017–18 revealed alarming statistics, including a total fertility rate (TFR) of 3.6, a stagnant modern contraceptive prevalence rate (mCPR) of 25% since 2012, and 17% of married women with unmet contraceptive needs [2]. These figures place Pakistan among those countries with relatively high fertility rates and substantial population growth trajectories, and thus highlight the urgent need for effective family planning (FP) interventions to improve reproductive health (RH) outcomes.

“Wanted fertility” refers to the desired number of children that couples or individuals wish to have. In Pakistan, the total wanted fertility rate is reported to be 2.9, indicating a desire for fewer children than the actual TFR. This suggests a level of unmet need for family planning services, as couples may desire fewer children than they currently have [2].

The unmet need for contraception represents the proportion of women who wish to avoid or delay pregnancy but are not using a contraceptive method. In Pakistan, 17% of married women are reported to have an unmet need for contraception [2]. This indicates a significant gap in the access to and utilization of family planning services, contributing to the high fertility rates and population growth.

Pakistan’s patriarchal norms and social system limits women’s power, agency, and economic opportunity. In Pakistan, daughters are often seen as an increased financial responsibility compared to sons due to the very low participation of daughters in the labor force, as well as the large costs incurred when daughters marry, due to dowries [3,4]. According to the Pakistan Labor Force Survey 2020–21, merely 16.8% of women engage in the labor force [5]. Women’s participation in productivity and financial empowerment is limited and primarily occurs within urban settings. As Pakistan’s rural population stands at 61 percent, most women rely on male members of the household for economic support and are less likely to contribute to household earnings [6].

Similarly, the dependence on sons is stronger in rural areas due to the association of agricultural work and land ownership with male inheritance; even in urban areas, boys are expected to carry the family name and care for their parents [7].

Studies have found that the preference for sons is a vital factor when deciding on permanent contraceptive methods in most of the Southeast Asian countries [8]. This preference is especially strong in India, Nepal, and Pakistan. In terms of temporary and traditional contraceptive methods, the study found a slight relation with son preference in this region [8].

Another study by Bongaarts (2013) explored sex preferences for male offspring, where Pakistan is ranked as second among the 61 countries [9]. These results on sex preference were also confirmed by the results of the series of Pakistan Health and Demographic Surveys (PDHSs) [9].

The prevailing gender preference for male offspring is a significant factor that influences FP practices in Pakistan and is deeply rooted in South Asian culture [10]. Societal views that sons provide financial stability and perpetuate family lineage foster a strong desire for male descendants. This preference has profound implications for FP programs, as couples may continue childbearing until a son is born, contributing to higher fertility rates and a lower contraceptive uptake. Gender preferences influencing FP decisions have been observed not only in Pakistan but also in other regions, particularly in Bangladesh and India [11,12]. In these regions, the desire for sons influences reproductive behavior, leading to differential, stopping behavior (DSB) or a male-preferring stopping rule. Understanding the impact of gender preferences on contraceptive uptake is critical for designing effective FP interventions tailored to the socio-cultural context of Pakistan.

Moreover, the persistence of gender-based discrimination and traditional patriarchal norms in Pakistani society exacerbates the challenges faced in implementing successful FP programs [13]. Women often face social and cultural barriers that limit their autonomy in reproductive decision-making, further complicating efforts to address unmet contraceptive needs and achieve the desired fertility rates. Despite efforts to promote FP and RH services in Pakistan, the prevalence of unmet contraceptive needs among married women remains high [2]. The persistence of gender preferences for male offspring exacerbates this issue, perpetuating high fertility rates and hindering contraceptive uptake. The societal emphasis on male descendants not only restricts women’s reproductive autonomy but also contributes to gender inequalities and perpetuates patriarchal norms [14]. Addressing the influence of gender preferences on contraceptive uptake is essential for improving reproductive health outcomes and advancing gender equality in Pakistan.

### Rationale

The rationale behind exploring the influence of gender preferences on contraceptive uptake among married women of reproductive age in Pakistan is multifaceted. Firstly, understanding the socio-cultural factors driving reproductive behavior is crucial for designing effective FP interventions that are tailored to the specific needs and beliefs of the population. Pakistan, like many other countries in South Asia, grapples with deeply entrenched gender norms and preferences, which significantly impact reproductive decision-making.

Secondly, addressing unmet contraceptive needs and promoting contraceptive uptake is vital for improving maternal and child health outcomes, reducing maternal mortality, and achieving sustainable population growth. By identifying barriers to contraceptive use, such as gender preferences for male offspring, policymakers and healthcare providers can develop targeted interventions to address these barriers and enhance access to family planning services.

Furthermore, addressing gender preferences for male offspring is essential for advancing gender equality and women’s empowerment in Pakistan. By challenging traditional patriarchal norms and promoting reproductive autonomy for women, efforts to improve contraceptive uptake can contribute to broader social and economic development goals, including poverty reduction and increased access to education and employment opportunities for women. In conclusion, examining the influence of gender preferences on contraceptive uptake in Pakistan is not only crucial for addressing unmet contraceptive needs and improving reproductive health outcomes, but also for promoting gender equality and women’s empowerment. By understanding the complex interplay between gender norms, societal expectations, and reproductive behavior, stakeholders can develop more effective strategies to promote family planning and advance the rights and well-being of women and girls in Pakistan. The objective of this analysis was to determine the association between the gender composition of children and modern contraceptive uptake among MWRA.

## 2. Materials and Methods

**Study Design:** The analysis in this paper is a secondary analysis of data collected from the “Naya Qadam” project (Naya Qadam means “a new step” in English). The Naya Qadam project focused on increasing contraceptive uptake among married women of reproductive age, specifically young women aged 15–24 years. The Naya Qadam project was launched by Pathfinder International in 2018 and concluded in 2021. Information in the “Naya Qadam” facility-based cross-sectional survey was collected between March and June 2019. A secondary analysis of the “Naya Qadam” dataset was used to study the association between gender composition and modern contraceptive uptake among MWRA in Pakistan.

**Study site:** The survey was carried out across three districts in the Sindh (Karachi, Shaheed Benizarabad, and Larkana) and three in the Punjab provinces (Okara, Pakpattan, and Rawalpindi).

**Sample Size Determination:** The “Naya Qadam” survey recruited a total of 1690 MWRA who visited healthcare facilities seeking health services. The sample size for the study was determined using the two-proportions population size formula. The parameters that were considered included design effect, intraclass correlation coefficient (ICC), proportions of contraceptive users, and refusal rate. The study aimed for a robust sample size, considering a design effect of 1.482, an ICC of 0.025, and a proportion of contraceptive users in Sindh and Punjab at 25% and 30%, respectively.

**Data Collection:** Trained data collectors conducted face-to-face interviews with MWRA who visited healthcare facilities seeking health services. A structured questionnaire was used to collect data on demographic characteristics, reproductive history, and contraceptive use.

**Questionnaire:** The questionnaire was developed based on the questionnaire used in the PDHS 2017–18 and consultations with experts in the field. It included validated measures to document the necessary information, ensuring the reliability and validity of the data collected.

**Data Management:** Data were collected on paper and were later transcribed into a secure electronic database. The data were cleaned to identify and resolve any inconsistencies or missing values. Quality control measures were implemented to ensure the accuracy and completeness of the data.

**Analysis:** The primary analysis focused on examining the association between gender composition and modern contraceptive uptake among MWRA. Logistic regression analysis was employed to estimate the adjusted odds ratios, considering potential confounding factors such as province, age, and health facility type. This statistical approach enabled the assessment of the strength and direction of the relationship between gender preferences and contraceptive behavior while controlling for covariates.

**Grouping for data analysis:** Among the interviewed MWRA, 495 participants, constituting 27% of the total sample, were included in this analysis. This subset was selected based on the criterion of adopting a family planning (FP) method, ensuring that the analysis focused specifically on modern contraceptive uptake among MWRA. To assess the effect of gender composition on modern contraceptive uptake, four binary categories were created based on the number of sons and daughters reported by the participants. The categories were designed to capture variations in contraceptive behavior based on the gender composition of children:
Category 1: group with zero sons and at least one or more daughters;Category 2: group with only one son and at least one or more daughters;Category 3: group with zero daughters and at least one or more sons;Category 4: group with only one daughter and at least one or more sons.


These categories allowed for a nuanced analysis of how the presence or absence of sons or daughters influenced modern contraceptive uptake among MWRA. A study conducted in Bangladesh also created similar categories of the gender composition of offspring to study the association between sex preference and contraceptive use [15].

**Ethical Considerations:** Ethical approval for the study was obtained from the Research and Development Solutions (RADS) institutional review board. Informed consent was obtained from all participants before data collection, and confidentiality and anonymity were maintained throughout the study. Participants were assured of their right to withdraw from the study at any time without repercussions.

By systematically implementing these methodological steps, the study ensured rigorous data collection, analysis, and interpretation, ultimately providing valuable insights into the complex interplay between gender composition and modern contraceptive uptake among MWRA in Pakistan.

## 3. Results

Table 1 presents participants’ demographic characteristics and contraceptive history. Of the 495 married women, the majority (68%) were aged 25–34 years old and had almost equal numbers of sons and daughters. Most women (96%) had used modern contraceptives at least once in their lives, whereas 38% were using modern contraceptives at the time of the interview. Among current modern contraceptive users, the pill was the most preferred method, followed by condoms (7%) and injections (6%).

The findings of the study revealed a significant association between offspring gender composition and modern contraceptive uptake among MWRA in Pakistan, highlighting the impact of male child preference on reproductive behavior. Table 2 presents the adjusted odds ratio (AOR) for all four categories, as well as their significance level. Regardless of province, age, and health facility type, the AOR for modern contraceptive uptake was notably low when the number of daughters increased without sons. Specifically, the AOR for modern contraceptive uptake in this scenario was 0.12 (95% CI: 0.04–0.34; *p* < 0.000), indicating a substantially decreased likelihood of adopting modern contraceptive methods among women with daughters but no sons.

Conversely, the presence of at least one son was strongly associated with increased modern contraceptive use among MWRA in Pakistan. Women who reported having at least one son exhibited a significantly higher likelihood of modern contraceptive uptake, with an AOR of 19.91 (95% CI: 8.00–49.50; *p* < 0.000). This finding underscores the influential role of son preference in shaping reproductive decision-making and contraceptive behavior among married women in Pakistan. In addition, when the number of daughters was fixed at zero and the number of sons increased, the adjusted odds of modern contraceptive usage demonstrated a gradual increase. Although this trend was not statistically significant at conventional levels (*p* = 0.061), the observed pattern suggests a potential association between son preference and modern contraceptive uptake, even in the absence of daughters.

Of particular note, the family composition of one daughter and two sons emerged as the most preferred scenario for modern contraceptive uptake among MWRA in Pakistan. Women in this category exhibited odds that were two times higher for modern contraceptive uptake compared to other gender compositions, highlighting the pronounced influence of gender preferences on reproductive behavior.

Overall, the results underscore the complex interplay between child gender composition and modern contraceptive uptake among MWRA in Pakistan. The findings emphasize the need for targeted interventions aimed at addressing gender biases and promoting informed, more autonomous decision-making regarding family planning.

## 4. Discussion

In Pakistan, as in many other Asian countries, there is a strong societal preference for male offspring. This preference is deeply rooted in cultural and socio-economic factors, including the desire for male heirs to inherit property and provide financial support to parents in old age [10]. The preference for sons can influence reproductive behavior, leading to higher fertility rates as couples may continue childbearing until they have a male child. Similar preferences for male offspring are observed across various Asian countries, particularly in South Asia, East Asia, and parts of the Middle East. However, the intensity of this preference may vary, with some countries experiencing more pronounced gender imbalances than others. For example, in countries like India and China, son preference has contributed to imbalanced sex ratios at birth, with a higher number of male births compared to female births [16]. Likewise, the sex preference of a child is a significant barrier to birth control in Nepal [17]. Healthcare providers in Karachi also confirmed that the preference for male offsprings is high in Pakistani society. Sons are preferred for their perceived social value, such as their role in continuing the family name; their inheritance rights; and their ability to support the family economically [7]. Another study conducted in Pakistan found that the preference for a specific sex, especially for sons, significantly hinders the use of contraceptive measures [18]. This phenomenon has implications for family structures, gender equality, and social stability, underscoring the need for targeted interventions to address underlying socio-cultural norms. In contrast, countries like Japan and South Korea have experienced shifts in gender preferences over time, with the increasing acceptance of gender equality and smaller family sizes.

Studies have shown that couples may continue childbearing until they have a male child, leading to higher fertility rates and lower contraceptive uptake [16]. Moreover, gender preferences can also affect the use of sex-selective technologies, such as sex-selective abortion or preimplantation genetic diagnosis, which may further skew sex ratios and impact reproductive health outcomes [13].

The findings of this study highlight the profound influence of gender composition on modern contraceptive uptake among married women of reproductive age (MWRA) in Pakistan. The strong societal desire for male offspring significantly shapes reproductive behavior, leading to differential contraceptive practices based on the gender composition of children. These findings corroborate previous research that demonstrates the pervasive role of son preference in perpetuating high fertility rates and low contraceptive uptake, particularly in patriarchal societies. The preference for sons not only perpetuates high fertility rates but also exacerbates gender inequalities and perpetuates patriarchal norms within Pakistani society. Women may face immense pressure to continue childbearing until a son is born, thereby limiting their autonomy and reproductive choices. This pressure reflects broader societal expectations and gender roles, which often prioritize the male child as the bearer of family lineage and the provider of financial stability.

Furthermore, the societal emphasis on male descendants undermines efforts to promote gender equality and women’s empowerment in Pakistan. The perpetuation of son preference fosters a cycle of gender-based discrimination, wherein women are relegated to subordinate roles within the family and society. This perpetuation of patriarchal norms not only restricts women’s reproductive autonomy but also hinders their ability to participate fully in social, economic, and political spheres. To address this issue, measures such as community-based awareness campaigns to challenge gender biases, economic empowerment programs to enhance women’s financial independence, and policy reforms to ensure equal inheritance rights for women can play a critical role. Additionally, integrating gender-sensitive curricula in schools and engaging religious and community leaders to advocate for gender equity can further help dismantle these entrenched societal norms. Moreover, efforts to promote gender equality and women’s empowerment must be integrated into broader development agendas, encompassing education, healthcare, economic opportunities, and social protection. By challenging entrenched patriarchal norms and promoting gender-sensitive policies and programs, Pakistan can move towards a more equitable and inclusive society where all individuals have the autonomy to make informed choices about their reproductive lives.

Addressing the root causes of gender preference and its impact on contraceptive uptake is essential for advancing reproductive health and gender equality in Pakistan. Effective interventions should focus on community engagement programs that challenge societal norms and beliefs surrounding gender roles, such as son preference, through dialogs led by influential local figures, including religious and community leaders. Promoting women’s autonomy in reproductive decision-making can be supported by initiatives like gender-sensitization workshops for couples and capacity-building programs that enhance women’s negotiation skills. Additionally, ensuring access to comprehensive family planning services tailored to the needs of MWRA, such as mobile health clinics in underserved areas and culturally sensitive counseling, can significantly improve contraceptive uptake and overall gender equity.

### Limitations

The survey was conducted in health facilities, which may not represent the broader population. Married women seeking health services may have different characteristics and behaviors compared to those who do not seek medical care, leading to a biased sample. In addition, the survey was limited to six districts in Pakistan, specifically three from each of the Sindh and Punjab provinces. These districts may not be fully representative of the cultural, socioeconomic, and geographic diversity present in the entire country, thus limiting the generalizability of the findings. Lastly, while the analysis adjusted for certain covariates such as province, age of women, and health facility type, other potential confounding variables, such as socioeconomic status, education level, and cultural factors, were not included, as these were not asked in the survey instrument. These unmeasured variables could impact contraceptive behavior and outcomes. Addressing these limitations would enhance the robustness and applicability of the study’s findings and recommendations.

## 5. Conclusions

In conclusion, the findings of this study highlight the significant influence of gender composition on modern contraceptive uptake among MWRA in Pakistan. The strong societal preference for sons drives reproductive behavior, leading to differential contraceptive practices based on the gender composition of children. Addressing gender norms and biases while promoting informed choice are essential steps toward enhancing modern contraceptive uptake and achieving sustainable population growth in Pakistan.

Effective interventions must challenge entrenched societal norms and beliefs surrounding gender roles and preferences. By promoting gender equality and women’s empowerment, individuals can make autonomous decisions regarding family planning, free from external pressures and constraints. Targeted strategies, including male engagement initiatives, are crucial for challenging patriarchal norms and fostering an environment where all individuals have the autonomy to make informed choices about their reproductive lives.

Furthermore, integrating gender-sensitive approaches into broader reproductive health programs and policies is essential for addressing the underlying determinants of gender preference and promoting equitable access to family planning services. By prioritizing gender equality and reproductive rights, Pakistan can create an enabling environment where individuals are empowered to make informed, self-determined decisions about their reproductive health, ultimately contributing to improved health outcomes and sustainable development for all.

## Figures and Tables

**Table 1 ijerph-22-00118-t001:** Participants’ demographic characteristics and contraceptive history.

Variable	Number of Women Using Modern Contraceptive	Percentage of Women Currently Using a Modern Contraceptive Method
**Province (*n* = 495)**		
Punjab	200	40%
Sindh	295	60%
**Health facility type (*n* = 495)**		
CMWs	117	24%
Public	222	45%
Private	156	32%
**Age categories (*n* = 495)**		
15–24 years	82	18%
25–34 years	309	68%
35+ years	63	14%
**Average no. of children (*n* = 495)**		
Daughter	2.0	
Son	1.8	
**Parity (*n* = 495)**		
1 Child	41	8%
2 Children	220	44%
3 Children	181	37%
>3 Children	53	11%
**Ever used (*n* = 495)**		
Yes	436	96%
No	18	04%
**Current use (*n* = 495)**		
Yes	173	38%
No	281	62%
**Current FP method mix (*n* = 173)**		
Pill	87	19%
Condom	32	7%
Injectable	28	6%
IUD	15	3%
Implant	6	1%
Female sterilization	5	1%

**Table 2 ijerph-22-00118-t002:** Analysis of gender composition and modern contraceptive uptake.

Categories	Number of Women Using Modern Contraceptive	Percentage of Women Currently Using a Modern Contraceptive Method	Adjusted Odd Ratio (95% CI)
**0 Son (*n* = 282)**			
**1 Daughter**	**41**	**39%**	**1**
2 Daughters	228	05%	0.13 *** (0.05–0.34)
3+ Daughters	13	79%	6.6 * (1.39–31.39)
**At least 1 Son (*n* = 402)**			
**0 Daughter**	**40**	**40%**	**1**
1 Daughter	220	21%	0.39 ** (0.19–0.78)
2 Daughters	93	62%	2.10 (0.93–4.71)
3+ Daughters	49	71%	3.60 ** (1.42–9.03)
**0 Daughter (*n* = 97)**			
**1 Son**	**40**	**40%**	**1**
2 Sons	30	87%	11.08 *** (2.72–45.11)
3+ Sons	27	100%	23.29 *** (4.13–131.32)
**At least 1 Daughter (*n* = 495)**			
**0 Son**	**41**	**39%**	**1**
1 Son	220	22%	0.45 * (0.22–0.93)
2 Sons	181	43%	2.11 * (1.01–4.39)
3+ Sons	53	72%	4.11 ** (1.64–10.27)

* *p* < 0.05; ** *p* < 0.01; *** *p* < 0.001.

## Data Availability

The raw data supporting the conclusions of this article will be made available by the authors upon request.

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
