# Peer review of "The Effect of Offspring Gender Composition on Modern Contraceptive Uptake Among Married Women of Reproductive Age in Pakistan: A Facility-Based Cross-Sectional Study"

_ijerph, 2025, doi:10.3390/ijerph22010118_

Round 1
Reviewer 1 Report
Comments and Suggestions for Authors
Excellent paper! It is well written and well organized. Good citation of cross-cultural research.
The abstract could be shortened.
How many people declined to participate in the study?
Other possible palliatives: urging women to delay marriage, especially to continue their education; provide government support to the aged, easing the burden on sons.
Line 83--offspring is the plural as well as the singular form.
Author Response
Comments 1: Excellent paper! It is well written and well organized. Good citation of cross-cultural research
Response 1: Thank you for your appreciation
Comments 2: The abstract could be shortened
Response 2: Thanks for pointing this out. We have removed the acronyms that are not required in the abstract. Moreover, the abstract has been further shortened where possible. You can see the changes with RED font and Strikethrough font.
Comments 3: How many people declined to participate in the study?
Response 3: A total of 24 people declined to participate in the primary study, which was a facility-based cross-sectional survey. This research article used the primary dataset of 1,690 married women of reproductive age to draw a subset of a sample (495) based on the research question and performed the necessary statistical analysis. This is already explained under the materials and methods section in the abstract.
Comments 4: Other possible palliatives: urging women to delay marriage, especially to continue their education; provide government support to the aged, easing the burden on sons.
Response 4: Unfortunately, we do not understand this comment. Can you please provide additional details and points of clarification?
Comments 5: Line 83--offspring is the plural as well as the singular form.
Response 5: Corrected.
Reviewer 2 Report
Comments and Suggestions for Authors
This manuscript describes the rates of contraceptive use in a specific population in Pakistan based on parity in a moderately sized sample. The study is interesting and innovative and well presented. The study is well designed and statistical methods appear appropriate.
Suggestion 1: While the results of the study are interesting, the conclusion that women prefer a specific family gender structure is not justified by this data alone. The conclusions throughout the manuscript would be revised to only state what the data can provide. For example line 29 "the gender composition of one daughter with two sons emerged as the most preferred gender composition..." is unfounded, but authors could state that this group had the highest contraceptive use "potentially because of" family composition preferences.
Suggestion 2: In the introduction their are some claims that need to be rephrased to remain neutral and scientific. Such as "Pakistan's patriarchal social system limits women's empowerment." While this may be true and supports the paper, could be said in a more scientific manner.
Suggestion 3: Table 1 and 2 need to be clarified. The total number of patients and the relative percentages in each category should be reported. The "%" column needs a more descriptive title as to not be misleading, such as "Percent of women actively using contraception". Table 2 should say "Atleast 1 Son" and "Atleast 1 daughter" in the header to be clear.
Suggestion 4: The organization of data is somewhat confusing and would potentially be better visualized with different groupings, however this is not necessary for publication. For example it would be nice to see the data represented by total parity in addition to presence of sons.
Suggestion 4: I would change the title to be more clear. "Gender composition" is somewhat confusing and something like "offspring gender composition" is more clear.
Author Response
Comment 1: This manuscript describes the rates of contraceptive use in a specific population in Pakistan based on parity in a moderately sized sample. The study is interesting and innovative and well presented. The study is well designed and statistical methods appear appropriate.
Response 1: Thanks for your overall comments and appreciation.
Comment 2: While the results of the study are interesting, the conclusion that women prefer a specific family gender structure is not justified by this data alone. The conclusions throughout the manuscript would be revised to only state what the data can provide. For example line 29 "the gender composition of one daughter with two sons emerged as the most preferred gender composition..." is unfounded, but authors could state that this group had the highest contraceptive use "potentially because of" family composition preferences.
Response 2: We have agreed with your comments/suggestions and the changes in line 29 have been done with RED font and Strikethrough font. In the conclusion section, we have highlighted the significance of gender composition on the uptake of modern contraceptive without stating or specifying the preferred family/gender structure.
Comment 3: In the introduction their are some claims that need to be rephrased to remain neutral and scientific. Such as "Pakistan's patriarchal social system limits women's empowerment." While this may be true and supports the paper, could be said in a more scientific manner.
Response 3: We have rephrased this sentence to state the following: Pakistan’s patriarchal norms and social system limits women’s power, agency, and economic opportunity.
Comment 4: Table 1 and 2 need to be clarified. The total number of patients and the relative percentages in each category should be reported. The "%" column needs a more descriptive title as to not be misleading, such as "Percent of women actively using contraception". Table 2 should say "Atleast 1 Son" and "Atleast 1 daughter" in the header to be clear.
Response 4: Thank you for highlighting this. The changes have been made accordingly.
Comment 5: The organization of data is somewhat confusing and would potentially be better visualized with different groupings, however this is not necessary for publication. For example it would be nice to see the data represented by total parity in addition to presence of sons.
Response 5: Thanks again. The average number of sons are mentioned in table 1. We have not mentioned parity by gender (son or daughter) to avoid long tables and repeated information.
Comment 6: I would change the title to be more clear. "Gender composition" is somewhat confusing and something like "offspring gender composition" is more clear.
Response 6. That is a good suggestion. We have added the word “offspring in the title”.
Reviewer 3 Report
Comments and Suggestions for Authors
The topic examined in this paper is important. The methodology is clearly explained, as are the results. The sample is substantial. The structure is clear and so is the discussion. However, the references are old and outdated: there are many recent publications on the issue of son preference, prenatal sex selection, contraception and the desire to have a son, which should be included.
Also, the discussion and conclusion is rather superficial: the authors talk about promoting gender equality and challenging societal norms, which seems quite vague. Please provide concrete examples of measures to address the issue examined.
Also, I would incorporate nuances in the discussion: were the women in the districts surveyed belonging to different social classes? It would be important to consider the factor of social class and the education of parents in promoting the practice of son preference, and to what degree it is impacted.
Also, the results show a preference for the woman taking full responsibility for contraception (30% the pill, IUD, implant, sterilization) vs the condom (only 7%), this should be discussed: are the husbands unwilling to share responsibility for not having children, even though certain contraceptives can have serious side effects on the wives?
Comments on the Quality of English LanguagePlease correct repetitions, for example line 88 and 280-281: perpetuates appears twice in the same sentence.
Author Response
Comment 1: The topic examined in this paper is important. The methodology is clearly explained, as are the results. The sample is substantial. The structure is clear and so is the discussion. However, the references are old and outdated: there are many recent publications on the issue of son preference, prenatal sex selection, contraception and the desire to have a son, which should be included.
Response 1. Thank you for your kind appreciation of the topic, methodology, results, and overall structure of the paper. We acknowledge the concern regarding the use of outdated references. To address this, we have incorporated the most recent available publications that discuss son preference, sex-imbalances, contraception, and the desire to have a son where applicable.
Comment 2: Also, the discussion and conclusion is rather superficial: the authors talk about promoting gender equality and challenging societal norms, which seems quite vague. Please provide concrete examples of measures to address the issue examined.
Response 2: We have suggested a few measures to address the said issue to make the discussion section stronger and clearer.
Comment 3: Also, I would incorporate nuances in the discussion: were the women in the districts surveyed belonging to different social classes? It would be important to consider the factor of social class and the education of parents in promoting the practice of son preference, and to what degree it is impacted.
Response 3: We have discussed certain limitations under the “Limitations” section, such as differences in the characteristics and behaviors of women who seek health services compared to those who do not. However, differences in social class and education are unlikely, as the survey was conducted at the same level of healthcare facilities (primary healthcare unit), and all participants were from rural areas of Pakistan.
Comment 4: Also, the results show a preference for the woman taking full responsibility for contraception (30% the pill, IUD, implant, sterilization) vs the condom (only 7%), this should be discussed: are the husbands unwilling to share responsibility for not having children, even though certain contraceptives can have serious side effects on the wives?
Response 4: In this study, the results indicated that the pill was the most preferred contraceptive method (19%), followed by condoms (7%) and injections (6%), while IUDs, implants, and sterilization were the least preferred options (refer to Table 1). According to the latest Pakistan Demographic and Health Survey 2017-18, condoms and female sterilization are the most commonly used methods (9% each), with injectables being the third most popular modern contraceptive method (3%). However, as this study was a facility-based cross-sectional survey, the generalizability of the findings is limited and this was discussed in detail under the “Limitation” section.